# DiffBody: Human Body Restoration by Imaging with Generative Diffusion Prior

## Abstract

Human body restoration is critical for a wide range of applications. Despite recent advances in general image restoration using generative models, their performance in human body restoration remains suboptimal, often resulting in noticeable artifacts, such as unnatural textures, misalignments that disrupt the structural integrity, and loss of fine body details. To address these challenges, we propose a novel approach by introducing a human body-aware diffusion model that leverages domain-specific knowledge to enhance restoration quality. Our method employs a two-stage diffusion-based image restoration model. In the first stage, we generate human body preliminary predictions such as normal and depth map (*priors*) from degraded images using a multi-channel joint diffusion model accompanied by a robust reconstruction paradigm. In the second stage, we reconstruct the restored image based on the priors generated in the first stage, while balancing the control strength of different priors to improve restoration quality. Extensive quantitative and qualitative experiments demonstrate the superiority of our approach in generating perceptually high-quality human body restoration results.

## 1 Introduction

Blind image restoration (BIR) aims to enhance the quality of degraded images through processes like denoising (Tian et al., 2020), sharpening (Wang et al., 2020), deblurring (Zhang et al., 2022), super-resolution (Liu et al., 2022), *etc.*, a domain that has seen significant progress with advancements in the data-driven learning paradigm. Although general BIR has made substantial strides, users often exhibit a greater interest in the specific effects of BIR on particular subjects, with the human body being one of the key focuses. The restoration of the human body can have a profound impact on various human-centric applications, such as improving portrait quality in social media apps and aiding related downstream tasks like person re-identification (Ye et al., 2021), 3D reconstruction (Wang et al., 2021a), *etc.*

Regarding the methodology of BIR, while the end-to-end reconstruction paradigm (Liang et al., 2021; Wang et al., 2021c) has made great progress, it struggles to handle complicated combinatorial and severe degradations. The generative paradigm offers a solution to this issue by harnessing the power of generative models, such as Generative adversarial networks (GANs) (Karras et al., 2018) and Diffusion models (Rombach et al., 2021). The priors of generative models possess a powerful "imagination" learned from large amounts of data, which can be used to fill in reasonable details to the degraded images. Thus, current diffusion-based image restoration models (Luo et al., 2023; Lin et al., 2023; Yu et al., 2024) have notably enhanced the perceptual quality and adaptability of restoration results, thereby expanding the applicability of image restoration in practical contexts.

Despite these advancements, the specific area of human body image restoration remains underdeveloped. It is worth noting that the theoretical upper bound of performance for human body restoration is arguably higher than that for general restoration, since existing knowledge of the human body can be utilized as *priors* to the restoration problem. However, current diffusion-based general restoration models (Yang et al., 2023; Lin et al., 2023; Yu et al., 2024) are prone to produce artifacts for low-quality human images, including unnatural textures and loss of fine body details, as illustrated in Figs. 1 and 2. This problem can be analyzed using the perception-distortion tradeoff (Blau & Michaeli, 2018): Although existing GANs and diffusion models successfully improve the image quality such that the output distribution is closer to nice-looking natural images, since humans are

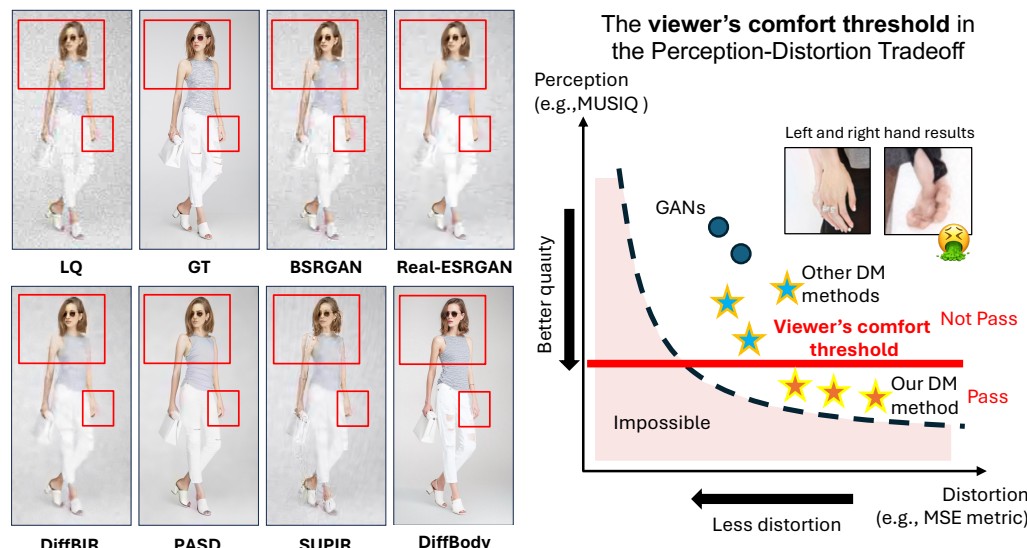

Figure 1: Human body image restoration method is required to produce an image with minimal distortion, high quality, and ensure viewer comfort, as humans are highly sensitive to distortions in limbs and skin. Our DiffBody model shows superior performance compared to other methods (left), particularly passing the viewer's comfort threshold (right).

extremely sensitive to distortions in limbs and skin, they have not reached the *viewer's comfort threshold*, resulting in uncomfortable user experience.

Our goal is to push the performance of human body image restoration beyond the viewer's comfort threshold. To this end, we present DiffBody, a novel and specialized two-stage diffusion model designed specifically for human body image restoration. The key idea is to smartly guide a pretrained diffusion model to restore clear and realistic human bodies through extracted human priors. In Stage 1, we use SwinIR (Liang et al., 2021) to preprocess the degraded image, following the approach of DiffBIR. This preprocessing generates a preliminary restoration, from which we extract key information such as pose and text. These elements, along with the preliminary restoration, are used to generate additional priors: a depth map, a normal map, and an improved preliminary restoration. These outputs provide critical color, structural and spatial guidance for the next stages of restoration. The depth map ensures structural alignment by accurately representing 3D shapes, while the normal map preserves surface details and corrects unnatural textures. Pose information maintains fine anatomical details and ensures overall human body visual coherence. In Stage 2, a detailed restoration is performed, where integrating multiple priors becomes crucial. Due to the complexity of inputs, an additional adapter is introduced to control color generation. Without it, inconsistencies in color and artifacts could undermine structural corrections. By incorporating the color adapter, we ensure consistent, accurate color, harmonizing structural and spatial details with precise color restoration. This integration significantly enhances the realism and quality of the restored images. While a formally-defined metric for quantifying the viewer's comfort threshold is not available, our user study show that the proposed method gives most viewer-comforting human body restoration as compared to existing methods.

Our main contributions are as follows: (1) Rather than forcing the model to strictly fit the low-quality distribution, we introduce a more flexible approach that allows the model's freedom in generation while guiding it to achieve the required *viewer's comfort threshold*. This enables better overall restoration performance, particularly in challenging human body restoration tasks; (2) We propose a novel two-stage framework. In Stage 1, we generate various priors from low-quality images to guide the restoration process. In Stage 2, these priors are leveraged to enhance human body image generation and restoration, exploring the impact of different types of priors on the final output quality; (3) We introduce an adapter module specifically designed to address color inconsistencies in the restoration process, ensuring accurate and realistic color reproduction in restored images.

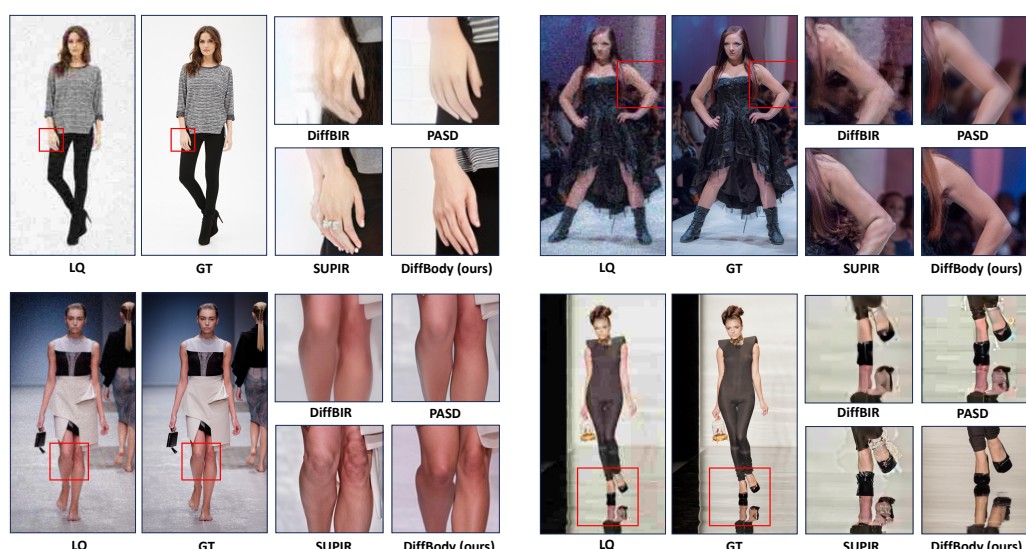

Figure 2: Our DiffBody model demonstrates superior performance on human body images compared to other state-of-the-art methods, particularly in terms of limb details, skin textures. Zoom in for better view.

## 2 RELATED WORK

**Perception-distortion tradeoff and evaluation methods:** (Blau & Michaeli, 2018) shows a tradeoff between perception and distortion: As the mean distortion (the dissimilarity to the ground truth image) decreases, the perceptual quality (the consistency with natural image statistics) must decrease as well. This tradeoff can be visualized as a distortion-quality curve (Fig. 1): Restoration results below this curve is impossible. Our goal is to push the performance below the viewer's comfort threshold on the perceptual quality, manifested in better perceived images with better perceptual metrics such as LPIPS (Zhang et al., 2018), ManIQA (Yang et al., 2022), ClipIQA (Wang et al., 2023), and MUSIQ (Ke et al., 2021), at the cost of potential visual distortion and lower objective metrics such as PSNR and SSIM. To assess viewer comfort, which cannot be measured by existing methods, we introduce the comfort pass test and comfort scoring in our user study.

**Blind image restoration:** Blind Image Restoration (BIR) aims to restore images without prior knowledge of the specific degradation model. Rather than relying on a known corruption process, BIR algorithms must generalize across different types of degradation, making it a more challenging task. Predominantly, existing literature (Bora et al., 2017; Menon et al., 2020; Daras et al., 2021; Pan et al., 2021; Yang et al., 2021b; Wang et al., 2021b) has concentrated on discerning a latent code situated in the latent space of a pre-trained GAN. Recent advancements in this domain (Ho et al., 2020; Song & Ermon, 2019; Song et al., 2020; Rombach et al., 2022; Ramesh et al., 2022; Saharia et al., 2022) have transitioned towards the utilization of DDPMs, marking a notable shift from conventional approaches. Other novel approaches such as DDRM (Kawar et al., 2022) utilizes SVD to address linear image restoration challenges, presenting an innovative and simplified approach. DDNM (Wang et al., 2022a) delves into vector range-null space decomposition to develop a novel sampling strategy, enhancing image restoration efficiency. DiffBIR (Lin et al., 2023) and SUPIR (Yu et al., 2024) aims to exploit a pretrained powerful generative prior to solve the BIR problem. In the realm of domain-specific image restoration models, a predominant emphasis has been placed on blind face restoration, as evidenced by works such as (Liu et al., 2022; Wang et al., 2022b; Gu et al., 2022). In contrast, the equally critical domain of human body restoration has not seen comparable development, a gap that our DiffBody model seeks to address.

**Controllable Human Image Generation:** Traditional methods for generating controllable human images mainly fall into two categories: those based on Generative Adversarial Networks (GANs) (Zhu et al., 2017; Siarohin et al., 2019) and those using Variational Autoencoders (VAEs) (Ren et al., 2020; Yang et al., 2021a), both leveraging reference images and specific conditions for input. Recent studies have ventured into enabling the generation process through textual instructions, though these tend to limit user input to basic pose or style adjustments (Roy et al., 2022; Jiang et al., 2022). State-of-the-art methods enable detailed control over vocabulary and pose including ControlNet(Zhang et al., 2023), T2I-Adapter(Mou et al., 2023), HumanSD(Ju et al., 2023), HyperHuman(Liu et al.,

2023), and CosmicMan (Li et al., 2024). These works have shown that diffusion models are capable to generate human images that contain rich detail and natural texture, which give us confidence that they can be utilized for high-quality human body image restoration.

## 3 METHODOLOGY

### 3.1 PRELIMINARY: LATENT DIFFUSION MODEL & STABLE DIFFUSION

Our exploration begins with the foundational principles of Latent Diffusion Models (LDM) (Rombach et al., 2022), which are pivotal in the generation of high-fidelity images from latent spaces. By compressing images into a lower-dimensional latent space before performing the diffusion process, LDMs achieve remarkable efficiency and detail in image synthesis. An autoencoder is used to transition between the image and its latent representation, effectively enabling the model to learn robust feature distributions.

Following the encoding phase, the model initiates a reverse diffusion process starting from a distribution of latent noise, gradually denoising this representation to reconstruct the image based on a given textual prompt. This process is facilitated by a U-Net architecture, which iteratively refines the latent features under the guidance of textual conditions embedded by a pre-trained text encoder such as CLIP. The primary objective in training these models involves minimizing the difference between the original and reconstructed images, formalized through a loss function that measures fidelity across multiple stages of the generative process.

### 3.2 DEGRADED IMAGE-DRIVEN JOINT DIFFUSION FOR HUMAN-CENTRIC PRIOR

In stage 1, the framework leverages degraded images as an integral component for generating human-centric priors in a diffusion process, as shown in Fig. 3 left. As illustrated in the model structure, degraded image $I_{LQ}$ is preprocessed by a robust image restoration model SwinIR (Liang et al., 2021) to produce preliminary restoration : $I_{ir} = \text{SwinIR}(I_{LQ})$. $I_{ir}$ is subsequently passed to MMPose(Sengupta et al., 2020) and LLaVA(Liu et al., 2024) to extract the human pose $I_{pose}$ and the corresponding textual prompt $p$, respectively: $I_{pose} = \text{MMPose}(I_{ir})$, $p = \text{LLaVA}(I_{ir})$. The prompt $p$ is then input into CLIP to extract the textual features $c_t = \text{CLIP}(p)$. With these foundational elements in place, we encode the latents of $I_{ir}$ and $I_{pose}$ using a VAE, producing $c_r = \mathcal{E}(I_{\text{res}})$ for the restored image and $c_p = \mathcal{E}(I_{pose})$ for the pose. $z_t$ and $c_p$ are then concatenated to form $\hat{z}_t$.

The initial training objective, guiding the first stage of model learning, is defined as:

$$L_{\text{U}} = \mathbb{E}_{z_t, t, c_t, c_p} \left[ \|\epsilon_d - \epsilon_{\theta_d}(\hat{z}_t, t, c_t)\|_2^2 + \|\epsilon_n - \epsilon_{\theta_n}(\hat{z}_t, t, c_t)\|_2^2 + \|\epsilon_i - \epsilon_{\theta_i}(\hat{z}_t, t, c_t)\|_2^2 \right]. \quad (1)$$

In this formulation, $\epsilon_d$, $\epsilon_n$, and $\epsilon_i$ represent three independently sampled Gaussian noise drawn from $\mathcal{N}(0, 1)$, for the depth, normal, and RGB branches. The terms $\epsilon_{\theta_d}$, $\epsilon_{\theta_n}$, and $\epsilon_{\theta_i}$ correspond to the three branches of the diffusion model, each tasked with predicting noise for the respective component. The multi-branch UNet is trained without the restored image latent $c_r$, allowing it to focus on generating the depth, normal, and RGB components based on the pose and textual conditions $c_t$ and $c_p$.

Once the UNet has been trained, we introduce the latent $c_r$ from the restored image and shift to training ControlNet (Zhang et al., 2023) with the following objective:

$$L_{\text{C}_1} = \mathbb{E}_{z_t, t, c_t, c_r, c_p} \left[ \|\epsilon_d - \epsilon_{\theta_c}(\hat{z}_t, t, c_t, c_r)\|_2^2 + \|\epsilon_n - \epsilon_{\theta_c}(\hat{z}_t, t, c_t, c_r)\|_2^2 + \|\epsilon_i - \epsilon_{\theta_c}(\hat{z}_t, t, c_t, c_r)\|_2^2 \right].$$
$$\quad (2)$$

In this phase, the ControlNet is trained with the full set of conditions including the restored image latent $c_r$, to refine the image restoration process by incorporating the prior from the higher-quality image. Stage 1 outputs three separate channels: $I_{res}, I_{depth}, I_{normal} = \mathcal{M}_1(I_{pose}, I_{ir}, c_t)$, which are then used in Stage 2 to further enhance the overall performance of human image restoration. The textual prompt is also updated in this stage, where $p' = \text{llava}(I_{res})$ is generated based on the refined image $I_{res}$.

### 3.3 ENHANCING HUMAN IMAGE RESTORATION THROUGH HUMAN-CENTRIC PRIOR

In Stage 2, with $I_{pose}$, $I_{depth}$, and $I_{normal}$ obtained from Stage 1, we utilize feature-extraction modules $\mathcal{F}_i$, which is built using convolutional neural networks (CNNs) and a fusion layer that combines these four priors, as shown in Fig. 3 right. The generative prior feature is computed as: $c_g = \alpha_1 \mathcal{F}_1(I_{ir}) + \alpha_2 \mathcal{F}_2(I_{pose}) + \alpha_3 \mathcal{F}_3(I_{depth}) + \alpha_4 \mathcal{F}_4(I_{normal})$. The restored image $I_{res}$ is first

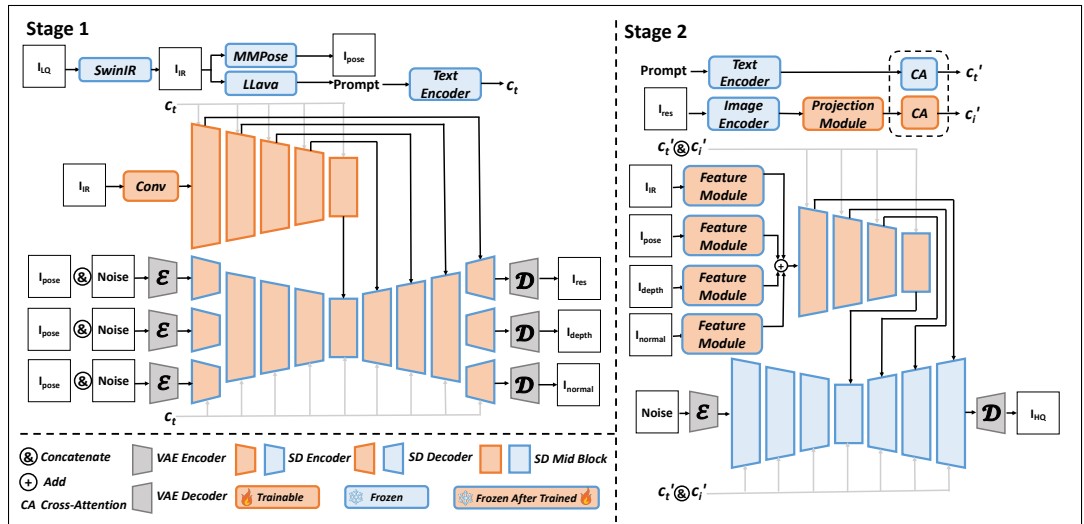

Figure 3: The workflow of the proposed DiffBody model.

encoded by CLIP and aligned using a projection module. After a cross-attention module, the prompt and $I_{res}$ are encoded as $c_i'$ and $c_t'$, respectively.

To generate the high-quality image $I_{HQ} = \mathcal{M}_2(I_{ir}, I_{pose}, I_{depth}, I_{normal}, c_t', c_i')$, the learning objective that guides our model training is defined as follows:

$$L_{C_2} = \mathbb{E}_{z_t, t, c_t', c_r, c_p} \left[ \left\| \epsilon - \epsilon_{\theta_c'}(z_t, t, c_t', c_g) \right\|_2^2 \right]. \tag{3}$$

In this formulation, $\epsilon$ represents a Gaussian noise term randomly extracted from $\mathcal{N}(0, 1)$, where $\epsilon_{\theta_c'}$ corresponds to the model's predicted noise for the given latent $z_t$ at time step $t$, conditioned on $c_t'$ and the generative prior $c_g$. Empirically, we find that providing $I_{ir}$ (the initial restoration) to the model, rather than $I_{res}$ (the further restored image), helps prevent the model from suffering from potential artifacts that may be introduced during Stage 1's restoration process, particularly when depth and normal maps are not yet available.

Once ControlNet has been trained, we introduce the latent $c_i'$ and train the color adapter using the full objective:

$$L_A = \mathbb{E}_{z_t, t, c_t, c_i, c_g} \left[ \left\| \epsilon - \epsilon_{\theta_c'}(z_t, t, c_t, c_i, c_g) \right\|_2^2 \right]. \tag{4}$$

In this training phase, fusing the $I_{res}$ information with the CLIP embedding broadens the model's learning paradigm to better capture color information. This fusion enables the model to handle color inconsistencies more effectively, resulting in more robust and higher-fidelity restoration. By integrating the degraded image with textual descriptions, poses, depth maps, and normal maps, our approach ensures a comprehensive restoration process, critical for recovering details lost to image degradation. This synergy of diverse inputs allows the model to restore images with greater accuracy, especially when critical information, such as color and fine details, has been obscured.

## 4 EXPERIMENTS

### 4.1 DATASETS

To address common challenges such as incomplete representations and variability in image quality, we implemented a comprehensive dataset annotation process, annotating each of the 5 million high-quality human images with MMPose, MiDaS depth (Ranftl et al., 2020), OmniNormal (Eftekhar et al., 2021), and LLaVA caption to create a robust and reliable training set. Using a bucket-based resizing strategy, similar to that in SDXL (Podell et al., 2023), we organized the dataset into five resolution buckets: 512×512, 512×768, 512×1024, 768×512, and 1024×512, ensuring the accommodation of varying resolutions. To maintain consistent quality across diverse image resolutions, we applied the degradation settings from Real-ESRGAN, simulating realistic image degradation.

The final training set includes approximately four million human images extracted and refined from the CosmicMan dataset (Li et al., 2024), which required croping and annotation to meet our training requirement.. Additionally, one million human images were sourced from various web-based repositories, providing broader diversity in poses and environments, with extensive filtering to meet our strict standards. For evaluation, we leveraged the SHHQ (Fu et al., 2022) dataset, a high-quality set of full-body human images, to serve as the test set in this paper, given its consistent image quality and resolution, making it a reliable benchmark for testing our diffusion model's capabilities.

### 4.2 EXPERIMENTAL DETAILS

For prior generation in stage 1, we employ Stable Diffusion 2.1-base as the foundational generative model. The three-branch architecture is initialized using the HumanSD framework, with fine-tuning applied only to the Stable Diffusion branch for 100,000 steps, using a batch size of 64. The model is optimized with the Adam optimizer at a learning rate of $10^{-5}$ and is conducted for one week on 8 NVIDIA A100 GPUs (80 GB). After this phase, the Stable Diffusion branch's parameters are frozen. The ControlNet branch, responsible for processing input $I_{ir}$, is then fine-tuned for another 100,000 steps, also with a batch size of 64. This second stage focuses on image restoration rather than general generation, and uses the same optimization settings and hardware.

For image restoration in stage 2, we use Stable Diffusion XL-1.0-base (SDXL) as the primary backbone. We initialize a trainable encoder block from SDXL and fine-tune it on features $I_{ir}$, $I_{pose}$, $I_{depth}$, and $I_{normal}$ over 100,000 steps, with a batch size of 32 and gradient accumulation of 2. This phase is optimized using Adam with a learning rate of $10^{-5}$ and takes approximately one week, utilizing 8 NVIDIA A100 GPUs. Following this, we initialize the color adapter with IP-adapterXL plus parameters and fine-tune it for an additional 200,000 steps with a batch size of 64. This final phase uses Adam with a learning rate of $10^{-4}$ and is trained under the same conditions and duration on 8 NVIDIA A100 GPUs. For inference, we utilize DDPM sampler (Ho et al., 2020) with 200 steps for both stage 1 and 2.

### 4.3 COMPARISONS WITH STATE-OF-THE-ART METHODS

**Evaluation Metrics.** In evaluating against ground truth, we utilize conventional metrics: PSNR, SSIM, and LPIPS (Zhang et al., 2018). To more accurately assess image authenticity for the BIR task, we incorporate non-reference image quality assessment (IQA) metrics: MANIQA (Yang et al., 2022), CLIPIQA (Wang et al., 2023), and MUSIQ (Ke et al., 2021) to enhance our evaluation framework. In the domain of human body restoration, we compare DiffBody with leading general image restoration methods: BSRGAN (Zhang et al., 2021), Real-ESRGAN+ (Wang et al., 2021c), Diff-BIR (Lin et al., 2023), PASD (Yang et al., 2023), and SUPIR (Yu et al., 2024). As shown in Table 1, DiffBody achieves strong performance on non-reference IQA metrics such as MANIQA, CLIPIQA, and MUSIQ. However, we observe relatively lower results on PSNR and SSIM. This aligns with findings in (Yu et al., 2024), which emphasize that traditional metrics like PSNR and SSIM are not highly indicative of true image quality in image restoration tasks. Fig. 6 and 5 show visual comparisons on the SHHQ dataset using the degradation method from the fifth row in Table 1. Additionally, Figures 4 present comparisons on real-world images from the Market1501 dataset, where no manual degradation was applied.

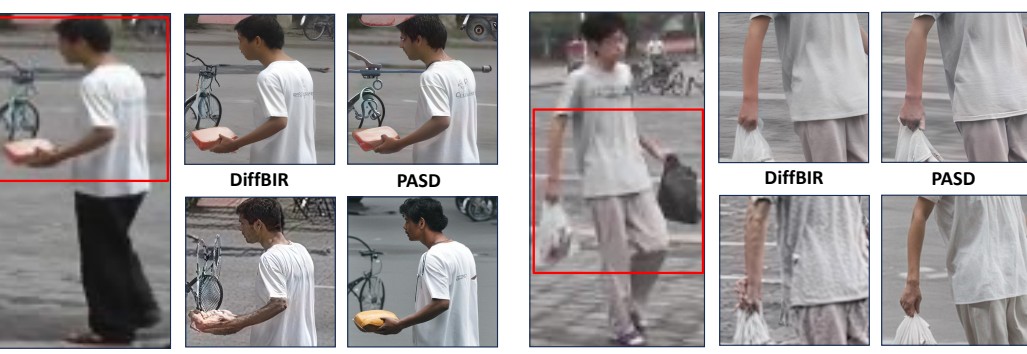

Figure 4: Qualitative comparison on real-world LQ images. Diffbody successfully recovers the human body details from $64 \times 128$ real-world LQ images.

Table 1: Quantitative comparison. Comparison of various methods across different degradation scenarios. green and blue represent the best and second-best performance, respectively. For metrics marked with ↓, lower values are better, while for the other metrics, higher means better.

| Degradation | Method | PSNR | SSIM | LPIPS↓ | ManIQA | ClipIQA | MUSIQ |
|---|---|---|---|---|---|---|---|
| **Mixture:** **Blur** ($\sigma = 2$) **SR** (×4) | BSRGAN | 32.42 | 0.7522 | 0.3604 | 0.3203 | 0.7329 | 58.0699 |
| | Real-ESRGAN | 31.08 | 0.7741 | 0.4944 | 0.1364 | 0.6234 | 15.0379 |
| | DiffBIR | 32.30 | 0.7368 | 0.3302 | 0.2918 | 0.7067 | 54.3575 |
| | PASD | 32.52 | 0.7637 | 0.2793 | 0.4029 | 0.7142 | 72.1634 |
| | SUPIR | 31.90 | 0.7143 | 0.2871 | 0.4475 | 0.7251 | 74.0450 |
| | DiffBody (ours) | 28.69 | 0.6423 | 0.1986 | 0.4532 | 0.7621 | 73.2073 |
| **Mixture:** **Noise** ($\sigma = 40$) **SR** (×4) | BSRGAN | 33.78 | 0.8400 | 0.1734 | 0.4548 | 0.7306 | 71.0124 |
| | Real-ESRGAN | 32.99 | 0.8428 | 0.1624 | 0.4235 | 0.5836 | 72.2913 |
| | DiffBIR | 34.15 | 0.8369 | 0.1610 | 0.3427 | 0.7156 | 69.6695 |
| | PASD | 33.31 | 0.7897 | 0.1733 | 0.4513 | 0.7224 | 75.6381 |
| | SUPIR | 33.55 | 0.7977 | 0.1633 | 0.4741 | 0.7250 | 75.0670 |
| | DiffBody (ours) | 29.36 | 0.6973 | 0.1973 | 0.4521 | 0.7421 | 76.3458 |
| **Mixture:** **Blur** ($\sigma = 2$) **Noise** ($\sigma = 40$) | BSRGAN | 31.04 | 0.7488 | 0.5071 | 0.2422 | 0.7120 | 18.7391 |
| | Real-ESRGAN | 30.87 | 0.7633 | 0.5341 | 0.2094 | 0.5984 | 14.3554 |
| | DiffBIR | 30.94 | 0.7104 | 0.4996 | 0.1794 | 0.6903 | 48.5516 |
| | PASD | 31.23 | 0.6897 | 0.5171 | 0.2607 | 0.6737 | 34.2320 |
| | SUPIR | 31.44 | 0.7028 | 0.3489 | 0.5103 | 0.7182 | 69.7255 |
| | DiffBody (ours) | 29.48 | 0.6327 | 0.1598 | 0.4494 | 0.7366 | 70.0132 |
| **Mixture:** **Blur** ($\sigma = 2$) **Noise** ($\sigma = 40$) **SR** (×4) | BSRGAN | 32.93 | 0.7997 | 0.2832 | 0.2355 | 0.7111 | 24.4447 |
| | Real-ESRGAN | 30.88 | 0.7665 | 0.5162 | 0.1707 | 0.5436 | 14.3322 |
| | DiffBIR | 31.65 | 0.7211 | 0.4493 | 0.2197 | 0.6960 | 60.2501 |
| | PASD | 31.85 | 0.7544 | 0.3470 | 0.4001 | 0.7022 | 56.8926 |
| | SUPIR | 31.50 | 0.7102 | 0.3474 | 0.4609 | 0.7131 | 66.0217 |
| | DiffBody (ours) | 29.86 | 0.6360 | 0.1360 | 0.4690 | 0.7405 | 68.8292 |
| **Mixture:** **Blur** ($\sigma = 2$) **Noise** ($\sigma = 20$) **SR** (×4) **JPEG** ($q = 50$) | BSRGAN | 32.93 | 0.7997 | 0.4800 | 0.3331 | 0.7150 | 58.9186 |
| | Real-ESRGAN | 31.55 | 0.7790 | 0.2719 | 0.3541 | 0.6011 | 61.0253 |
| | DiffBIR | 33.03 | 0.7879 | 0.2622 | 0.3427 | 0.7043 | 62.4461 |
| | PASD | 32.79 | 0.7854 | 0.2117 | 0.4019 | 0.7208 | 74.1890 |
| | SUPIR | 32.37 | 0.7533 | 0.2334 | 0.4780 | 0.7231 | 74.4595 |
| | DiffBody (ours) | 30.11 | 0.7202 | 0.1402 | 0.4861 | 0.7561 | 75.7115 |

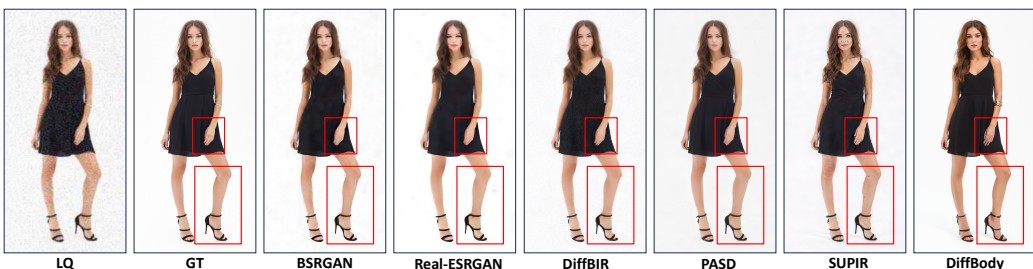

LQ  GT  BSRGAN  Real-ESRGAN  DiffBIR  PASD  SUPIR  DiffBody

Figure 5: Qualitative Comparison with different methods. Our model is more effective in generating detailed limbs and natural skin texture.

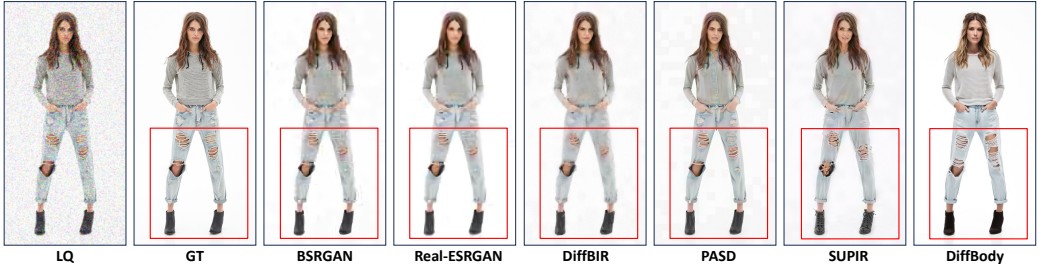

LQ  GT  BSRGAN  Real-ESRGAN  DiffBIR  PASD  SUPIR  DiffBody

Figure 6: Qualitative Comparison with different methods. Our model is more effective in generating natural texture and maintaining overall human body visual quality.

## 4.4 ABLATION STUDIES

**Effectiveness of LQ-Image Arrangement in Joint Diffusion:** We evaluate the effectiveness of different ways of arranging the low-quality (LQ) image input within the joint diffusion framework by comparing three methods. The first method, LQ Only, uses only the low-quality image as input to ControlNet, without pose information, serving as a baseline to assess image restoration based solely on the low-quality input. The second method, LQ+Pose, feeds both pose and low-quality signals into ControlNet to explore how conditioning on both inputs affects restoration performance. In the third method, LQ+Pose2U, the low-quality image is provided to ControlNet while the pose information is fed directly into the UNet, allowing us to assess the impact of splitting the conditioning between the ControlNet and the UNet. These methods are compared to determine the most effective conditioning strategy for image restoration. For quantitative analysis, we calculate the $L_2$ loss between the generated depth / normal maps with directly inferecing the depth / normal maps from the ground truth high quality image as shown in Table 2 and 3. Visual examples can be seen in Fig. 7 and Fig. 8. The depth and normal maps generated by method 3 are the closest to the ground truth. For clearer visualization, we also provide a relative heatmap that highlights the differences between the generated maps and the ground truth.

Table 2: $L_2$ loss comparison of depth map.

| Method | $L_2^d$ |
|---|---|
| LQ Only | 531.2 |
| LQ+Pose | 561.8 |
| LQ+Pose2U | 488.7 |

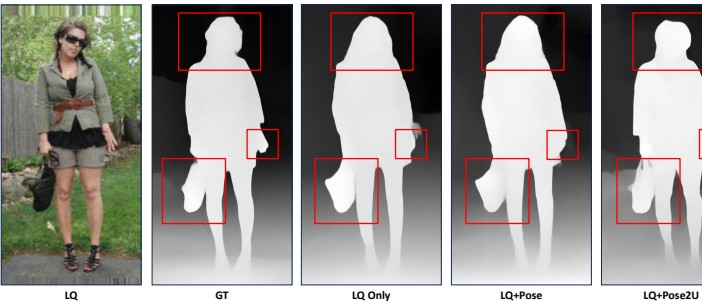

Figure 7: Visual Comparison of depth map.

Table 3: $L_2$ loss comparison of normal map.

| Mode | $L_2^n$ |
|---|---|
| LQ Only | 151.9 |
| LQ+Pose | 180.2 |
| LQ+Pose2U | 106.8 |

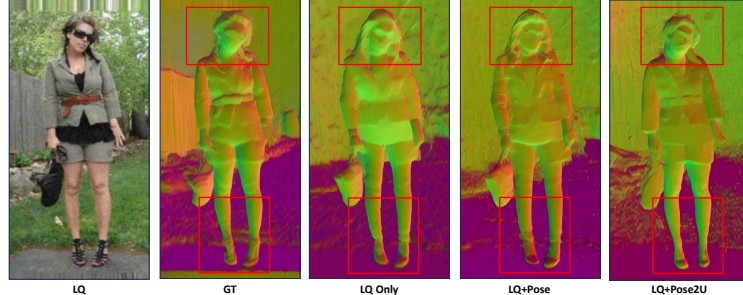

Figure 8: Visual Comparison of normal map.

**Effectiveness of Different Priors:** Then we compare the effectiveness of the three generative priors—depth, normal, and pose—used in our model. To assess their individual contributions, we trained three separate models, each excluding one of the priors (without pose, without depth, and without normal), and compared their performance against our full model, which incorporates all three priors. The results, as shown in Table 4, provide insight into how each prior affects the image restoration quality across several metrics. Our full model, leveraging all three priors, achieves the best overall performance, demonstrating the critical role of combining pose, depth, and normal priors for improved restoration results. Visual comparisons of the different models are provided in Figures 9, illustrating the qualitative impact of each prior on the restoration process.

Table 4: Quantitative comparisons. Notations follow those in Table 1. The model utilizing all priors achieves the overall best results, demonstrating the effectiveness of incorporating multiple priors.

| Depth | Normal | Pose | PSNR | SSIM | LPIPS ↓ | ManIQA | ClipIQA | MUSIQ |
|---|---|---|---|---|---|---|---|---|
| ✓ | ✓ |  | 28.72 | 0.7265 | 0.1907 | 0.4394 | 0.7603 | 73.7625 |
| ✓ |  | ✓ | 30.25 | 0.7243 | 0.1986 | 0.4436 | 0.7498 | 71.0442 |
|  | ✓ | ✓ | 28.11 | 0.6924 | 0.2105 | 0.4332 | 0.7492 | 70.8363 |
| ✓ | ✓ | ✓ | 30.11 | 0.7402 | 0.1402 | 0.4861 | 0.7561 | 75.7115 |

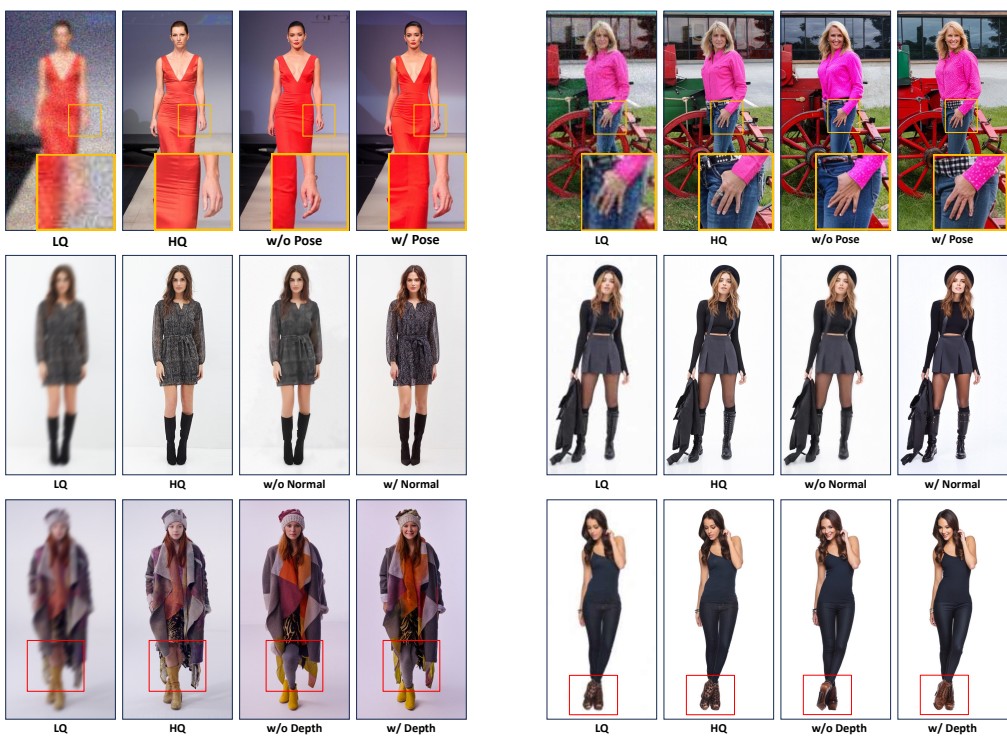

Figure 9: Qualitative comparisons: First Row: Comparison with and without pose information. Incorporating pose leads to improved limb details. Second Row: Comparison with and without the normal map. Incorporating the normal map improves human skin textures. Third Row: Comparison with and without depth information. Incorporating depth improves 3D spatial relationships in the generated images.

**Effectiveness of Color-controlling Adapter:** Finally, we evaluate the impact of incorporating the color-controlling adapter (color-Ada) by comparing model performance with and without the adapter. Since PSNR and SSIM are not well-suited for measuring color information in RGB images, we instead use CPSNR (Color Peak Signal-to-Noise Ratio) and CSSIM (Color Structural Similarity Index). CPSNR extends PSNR by accounting for color channels, allowing for a more accurate assessment of color fidelity. Similarly, CSSIM is a variant of SSIM that measures structural similarity across the color channels, providing a better evaluation of color consistency. The results, as presented in Table 5, demonstrate a significant improvement in performance when the color adapter is utilized. Visual examples of this comparison are provided in Fig. 10, further illustrating the qualitative improvements introduced by the color-controlling adapter.

Table 5: Quantitative comparison. The color adapter improves all numerical metrics, demonstrating its effectiveness in enhancing the image restoration process.

| Method | CPSNR | CD-SSIM | LPIPS ↓ | ManIQA | ClipIQA | MUSIQ |
|---|---|---|---|---|---|---|
| w/o color-Ada | 24.31 | 0.6423 | 0.1872 | 0.5160 | 0.7410 | 72.9950 |
| w/ color-Ada | 29.12 | 0.6821 | 0.1402 | 0.5380 | 0.7561 | 75.7115 |

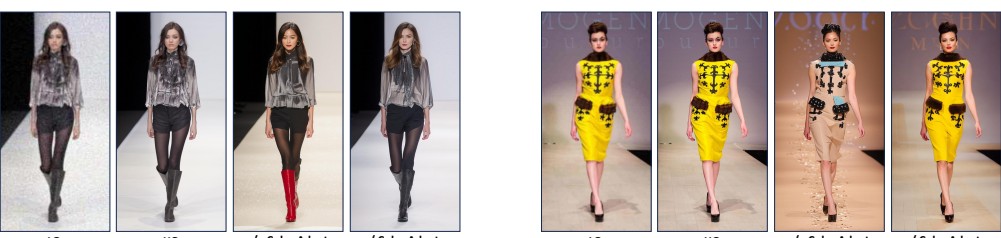

Figure 10: Qualitative comparison with and without the color adapter. The results show that incorporating the color adapter significantly enhances fidelity and overall visual quality.

### 4.5 USER STUDY

We conducted a user study to assess whether a method passes the viewer comfort test, as metrics like PSNR, LPIPS, or ManIQA cannot evaluate this aspect. We processed 50 low-quality human body images using six methods, including ours, and presented them to 10 volunteers. They answered three questions: (1) "Do you feel comfortable looking at this image?" (a yes/no comfort test), (2) "Can you rate your comfort level from 0 to 10?" (a continuous scale), and (3) "Select the best output from the six methods by evaluating each one based on its fidelity to the input image, overall quality, and viewer's comfort level." (a performance selection question). The results are shown in Fig. 11.

Since GAN artifacts (e.g., poor quality, lack of detail) differ from diffusion models, we only present results from four diffusion-based methods for the first two questions. Our method achieved the highest comfort pass rate (81.25%) and comfort score (7.53), outperforming other models. For the performance selection question, our method was preferred by users, with a selection rate of 58.32%.

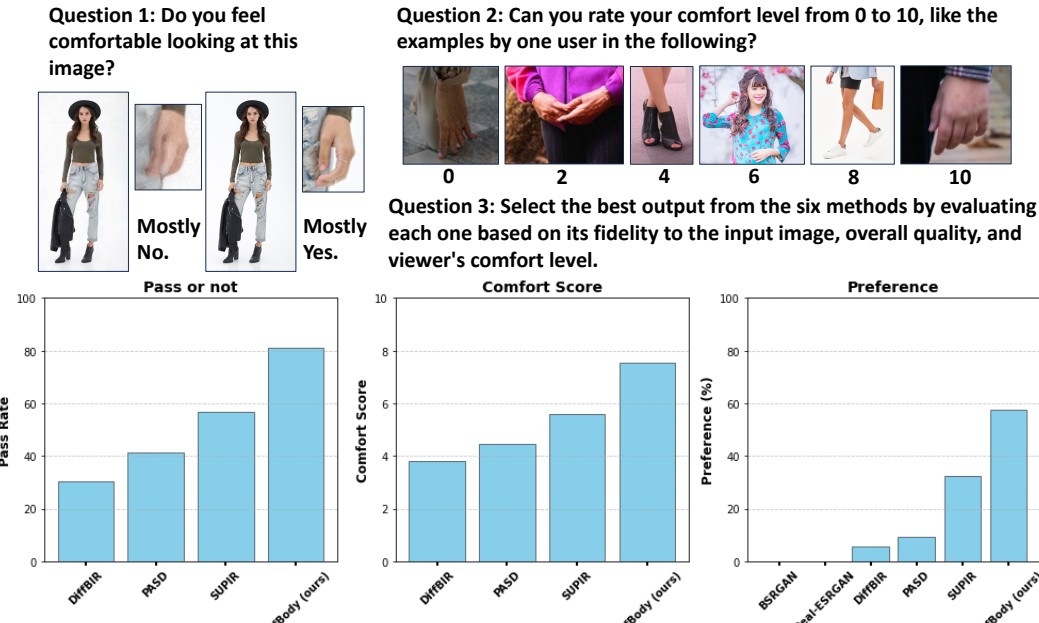

Figure 11: User study. Questions and example answers are shown on the top, while the results are shown on the bottom, including the viewer comfort pass test, comfort level scoring, and overall preference. The results clearly demonstrate that our method significantly outperforms the others.

## 5 CONCLUSIONS

DiffBody introduces a novel framework for human body restoration, achieving realistic outcomes by incorporating human-centric guidance into the pre-trained Stable Diffusion model. By leveraging various human-specific conditions, we surpass the capabilities of existing general image restoration models in addressing artifacts. A key aspect of our approach is balancing different priors, such as pose, depth, and normal maps, to strike a balance between the viewer's comfort threshold and fidelity to the low-quality (LQ) image. However, there are still areas for improvement, such as exploring advanced techniques like mesh modeling for precise body structure manipulation and ensuring the preservation of personal identity throughout restoration. Future work will focus on handling more challenging scenarios, including complex poses, multi-human images, and cases where subjects are partially occluded by objects. These extensions, along with better body control and identity preservation, will further enhance the robustness and applicability of human image restoration models.

**Ethical concerns:** While DiffBody offers significant advancements in human body restoration, it raises ethical concerns related to privacy, consent, and image counterfeiting. The ability to manipulate and restore human images could lead to unwanted alterations of an individual's likeness, potentially infringing on personal rights. Misuse of this technology may result in unauthorized modifications or counterfeit images. It is essential that this model is applied responsibly, with explicit consent, and that strong safeguards are in place to prevent misuse. Developers and researchers must remain vigilant in addressing these ethical challenges.

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
