# OpenReview forum: "DiffBody: Human Body Restoration by Imagining with Generative Diffusion Prior"
_ICLR.cc/2025/Conference — ICLR 2025 Conference Withdrawn Submission_

### Official Review · Reviewer_T2hz · 2024-10-28

**Soundness:** 2
**Presentation:** 2
**Contribution:** 2
**Rating:** 3
**Confidence:** 4

**Summary:**

A approach is proposed in this study, which introduces a human body-aware diffusion model to enhance the quality of restoration.

The model leverages domain-specific knowledge to achieve this goal.

In the first stage, a multi-channel joint diffusion model is employed to generate preliminary predictions of the human body, such as normal and depth maps (referred to as priors), from degraded images.

This process is accompanied by a robust reconstruction paradigm.

In the second stage, the restored image is reconstructed based on the priors generated in the first stage. The control strength of different priors is carefully balanced to improve the quality of restoration.

Extensive quantitative and qualitative experiments have been conducted to evaluate the performance of our approach. The results demonstrate the superiority of our method in generating perceptually high-quality restoration outcomes for human body images.

**Strengths:**

Quantitative and qualitative experiments show its superiority of the proposed methods.

**Weaknesses:**

Based on the description provided, it appears that the novelty of the research paper is somewhat limited. The content seems to align more with a technique paper focused on proposing training methods rather than presenting groundbreaking research.

**Questions:**

see in weakness

---

### Official Review · Reviewer_K6uB · 2024-10-29

**Soundness:** 3
**Presentation:** 2
**Contribution:** 3
**Rating:** 5
**Confidence:** 4

**Summary:**

This paper presents a human body restoration method that leverages priors from MMpose, LLaVA, and SwinIR. To effectively use these priors, a two-stage training approach is employed: in the first stage, a multi-branch U-Net is trained to predict the restored image, depth, and normal maps; in the second stage, a ControlNet model is trained, conditioned on these priors. For evaluation, a dataset of 5 million human images is collected. Visualization results demonstrate the superiority of the proposed method.

**Strengths:**

1. The proposed method is robust, showing significant improvements in visual results.
2. The evaluations are comprehensive, encompassing both qualitative and quantitative analyses.

**Weaknesses:**

1. The methodology section lacks clarity. For instance, in Lines 186–191, what does I_res represent, and how is z_t derived? Additionally, in Figure 3, is it correct to show three instances of I_pose?
2. The model’s size is not specified. How many U-Nets are utilized for training?
3. The quantitative metrics (PSNR and SSIM) are low; could you provide some explanation? Also, how does the method perform with SR (X2)? Since in SR (X4), the high-resolution images predicted by DiffBody show lower PSNR and SSIM yet appear plausible compared to the low-resolution input, SR (X2) might retain more detail and help assess the proposed method’s fidelity to the low-resolution input.
4. Will the collected 5M dataset be released?

**Questions:**

While the visual results are highly inspiring, the methodology section lacks clarity, making it challenging to fully understand the implementation details. Additionally, it is recommended that the authors release the code and dataset, as this would significantly benefit the community and enable further research in this area.

**Details Of Ethics Concerns:**

The paper introduces a new dataset of human images, which could raise potential privacy concerns.

---

### Official Review · Reviewer_AP4E · 2024-11-01

**Soundness:** 2
**Presentation:** 3
**Contribution:** 3
**Rating:** 5
**Confidence:** 4

**Summary:**

This paper propose a two stage method for human body restoration task. In the first stage, priors including normal and depth map are generated by a joint diffusion model. In the second stage, image is reconstructed based on the priors.

**Strengths:**

1. The visualization and quantification results demonstrate that this work has advantages over previous methods.

**Weaknesses:**

1. Some details in this paper are not well explained, which can sometimes be confusing for readers. These issues are listed in the "Questions" section.

**Questions:**

1. In line 236, is there a mistake in the correspondence between prompt, Ires, and ci, ct?
2. In stage2, Iir instead of Ires is provided to extract image feature and it is claimed to prevent the model from suffering from potential artifacts, so why not use the original low quality image?
3. Color adapter is an important component in DiffBody, however, color adapter is not found in Figure3 and it is unclear how color adapter works.
4. In Figure10, the boot of w/o Color Adapter is red, even without color adapter, the feature from Iir is so weak to change black to red?
5. In Figure10, the right leg of w/o Color Adapter is in the front, while in w/ Color Adapter, left leg is in the front. Is pose used in the result of  w/o Color Adapter? Is the Color Adapter able to correct pose？
6. In Figure3, when training unet of stage1, is the noise in fact Ires/Idepth/Inormal plus noise?

---

### Official Review · Reviewer_acnn · 2024-11-02

**Soundness:** 2
**Presentation:** 3
**Contribution:** 2
**Rating:** 3
**Confidence:** 4

**Summary:**

The authors propose a two-stage framework tailored for human body restoration. In Stage 1, the authors apply SwinIR for initial image enhancement and generates additional priors such as depth and normal maps, which help capture 3D structure and surface detail. Stage 2 then employs a diffusion model, guided by ControlNet, to integrate these priors and refine the output, ensuring a high level of realism. To manage color consistency across different priors, a color adapter is utilized. Quantitative and qualitative evaluations show that the method performs well in realism and detail preservation for human body restoration.

**Strengths:**

The visual quality of the restored images look good.

The user study looks comprehensive along with the user comfort level metric.

**Weaknesses:**

The use of notations is cumbersome and not well-defined. See specific instances in the questions section.

The entire methods section is cumbersome and appears to be a combination of multiple existing methods, lacking a proper ablation study to support these combinations. For example, there should be an ablation study demonstrating how the restored images appear at each stage, as well as one for the sub-stages within each stage. Currently, there is insufficient evidence to indicate which steps are beneficial and which are not. Without this, the paper reads as a collection of claims without proper justification.

In addition to the stages, there are certain claims in the paper that lack theoretical or empirical justification (e.g., lines 245-248, 253-254). Please refer to the quotations section.

It also appears that in Figures 5 and 6, identity and hairstyle are altered. More qualitative results are needed for a thorough evaluation.

**Questions:**

In line 190, should it be c_t instead of z_t​? There is no prior mention of z_t​, or is z_t​ the latent representation of I_{LQ}​?

Where are the functions M_1​ and M_2​ defined, and what do they represent?

Why is I_{ir}​ used instead of I_{res}​ for the initial training of the second stage (lines 245-248)? Where is the empirical evidence supporting this choice? An ablation study on this would be beneficial.

Where is the theoretical or empirical evidence showing that fusing I_{res}​ information with CLIP embeddings broadens the model’s learning paradigm (lines 253-254)?

What exactly is referred to as the "color adapter"? Is this the same ControlNet model trained in the second stage? This is not clearly defined.

What is the likelihood of overlapping images between the created training set and the SHHQ evaluation set? Were appropriate measures taken to prevent this overlap?

---

### Note · Authors · 2024-11-13

**Comment:**

I have read and agree with the venue's withdrawal policy on behalf of myself and my co-authors.

**Withdrawal Confirmation:**

I have read and agree with the venue's withdrawal policy on behalf of myself and my co-authors.